# Diversity Begets Diversity When Diet Drives Snake Venom Evolution, but Evenness Rather Than Richness Is What Counts

**DOI:** 10.3390/toxins15040251

**Published:** 2023-03-29

**Authors:** Romane Schaeffer, Victoria J. Pascolutti, Timothy N. W. Jackson, Kevin Arbuckle

**Affiliations:** 1Département Biologie and Geosciences, Faculté Sciences et Ingénierie, Université Toulouse III—Paul Sabatier, 31062 Toulouse, France; 2Department of Biosciences, Faculty of Science and Engineering, Swansea University, Swansea SA2 8PP, UK; 3Australian Venom Research Unit, Department of Biochemistry and Pharmacology, Faculty of Medicine, Dentistry and Health Sciences, University of Melbourne, Melbourne, VIC 3010, Australia

**Keywords:** venom evolution, toxin diversity, diet breadth, snake venom ecology, phylogenetic comparative analysis

## Abstract

Snake venoms are primarily used to subjugate prey, and consequently, their evolution has been shown to be predominantly driven by diet-related selection pressure. Venoms tend to be more lethal to prey than non-prey species (except in cases of toxin resistance), prey-specific toxins have been identified, and preliminary work has demonstrated an association between the diversity of diet classes and that of toxicological activities of whole venom. However, venoms are complex mixtures of many toxins, and it remains unclear how toxin diversity is driven by diet. Prey-specific toxins do not encompass the molecular diversity of venoms, and whole venom effects could be driven by one, few, or all components, so the link between diet and venom diversity remains minimally understood. Here, we collated a database of venom composition and diet records and used a combination of phylogenetic comparative methods and two quantitative diversity indices to investigate whether and how diet diversity relates to the toxin diversity of snake venoms. We reveal that venom diversity is negatively related to diet diversity using Shannon’s index but positively related using Simpson’s index. Since Shannon’s index predominantly considers the number of prey/toxins, whereas Simpson’s index more strongly reflects evenness, we provide insights into how the diet–venom diversity link is driven. Specifically, species with low diet diversity tend to have venoms dominated by a few abundant (possibly specialised) toxin families, whereas species with diverse diets tend to ‘hedge their bets’ by having venoms with a more even composition of different toxin classes.

## 1. Introduction

Predator–prey interactions have long been proposed to be important drivers of both lineage and phenotypic biodiversity due to antagonistic coevolution [1]. Snake venom has proven to be an excellent model system for this area as it enables clear links to be drawn between genotype and ecological function (prey subjugation and defence) [2,3]. One of the most apparent features of most snake venoms is their ‘complexity’, usually taken to mean their diverse complement of toxins, leading to their frequent description as ‘complex cocktails’ [4]. The primary function of snake venom is prey subjugation, so understanding variation in the composition of venoms has focused on the relationship between whole venoms or their component toxins and prey [4]. Snakes exhibit substantial diversity in their diets, both between and within species, which, in combination with the diversity in venom composition, provides a powerful system to understand how the biodiversity of predator traits is impacted by the diversity of their prey [2].

The importance of diet to the evolution of snake venom is well established. For instance, in a now-classic study, Daltry et al. [5] demonstrated that variation in venom composition amongst populations of *Calloselasma rhodostoma* could not be explained by either geographic or genetic distance but was well explained by dietary variation. Similarly, ontogenetic shifts in venom composition and activities are related to ontogenetic shifts in feeding ecology in Australian elapids [6,7]. Whole venom lethality is also higher to natural prey items than to non-prey species in *Echis* species [8,9], demonstrating that the association between diet and venom composition can have high ecological specificity. The loss of toxicity coincident with a shift to prey that does not require subjugation, for instance, in egg-eating sea snakes [10], or that can be subjugated by other means such as constriction [11] also adds evidence that diet is the primary driver of snake venom evolution.

In addition to studies considering whole venom, either in terms of composition or function, the existence of toxins which exhibit specificity in potency towards particular prey taxa further speaks to the role of diet in snake venom evolution. For instance, the venoms of species of *Boiga* and *Oxybelis* contain toxins which are potent against birds and/or lizards (the largest part of their diet) but have very low toxicity towards mammals [12,13,14]. A recent example demonstrated two separate prey-specific three-finger toxins in the venom of a single species: *Spilotes sulphureus* contains one toxin which is highly toxic to lizards but not mammals, and another toxin which exhibits the opposite pattern [15].

Covariation between prey types and venom lethality and/or toxin types may demonstrate that diet is a driver of snake venom evolution, but it does not provide a satisfying understanding as to why venom compositions are (often) so diverse. Prey-specific toxins differentially affecting taxonomic classes such as mammals and birds suggest that just a few related toxins could be sufficient to subdue most natural prey. For instance, in the example of *Spilotes sulphureus*, the two toxins identified are from a single protein family, potentially cover many species of prey, and are highly abundant in the venom [15]. Similarly, higher toxicity of the whole venom to natural prey could, in principle, be driven by many, few, or a single toxin. In both cases, there is no particular reason to suspect selection for highly diverse venoms and, in fact, a recent comparative study found that prey-specific venom toxicity is largely driven by dietary specialists [16]. These results, taken together, suggest that prey-specific venom attributes may be prone to lead to specialization and simplification of venom, and we need another way to explain diverse venom compositions. Evolutionary arms races provide one good solution, as they predict the evolution of multiple mechanisms to incapacitate prey [4], but there is conflicting evidence for this, and promising examples often arise from within a toxin class [17], and so may not explain the diversity of toxin families (but see discussion in [7]).

Rather than focusing on specific interactions between venoms or their components and particular prey taxa, some recent work has focused on the diversity (rather than the specific composition) of diet and venom toxins. Individual realisations of particular toxin/venom–diet relationships and mechanisms (broad-spectrum or prey-specific toxins) may vary greatly [7], but underlying generalisations and principles may arise from a consideration of the diversity itself. Therefore, it is reasonable to hypothesise that venom diversity should be driven by prey diversity. Consistent with this, preliminary work by Davies and Arbuckle [18], which focused on classes of toxicological activity rather than toxin composition, identified limited relationships between individual prey categories and venom activities but did find that more diverse diets were associated with more toxicologically diverse venoms. Their proposed explanation was that more generalist diets present a wider variety of physiological targets in natural prey which should select for diverse toxic actions to incapacitate such a range of prey. However, their broad, binary, and human-centric coding of diet and venom activities (based on a clinical database), left much scope for more quantitative and fine-scaled studies using less noisy data capable of distinguishing between two important aspects of biodiversity: richness and abundance.

Biological diversity indices are used in ecological research to quantify the diversity of species in a given area [19] and incorporate both richness (traditionally of species) and abundance of each species to varying extents. Diet diversity can be readily quantified using such indices for the number of prey species and their abundance in diet records for a given snake species. Venomic data on the toxin composition of snake venoms enables quantitative measures of both richness (number of protein families) and abundance (relative abundance of each family in the venom) such that ecological diversity indices can be used to quantify toxin diversity in venoms in the same way. Two commonly used measures, Shannon’s and Simpson’s diversity indices, quantify biodiversity in informatively distinct ways [19]. In particular, Shannon’s index puts more weight on the richness component (perhaps closer to what many non-ecologists think when they hear the word ‘biodiversity’), whereas Simpson’s index puts more emphasis on the abundance components (such that higher values are associated with more even compositions less dominated by a few types).

Recently, Holding et al. [20] showed that snake venom complexity evolves alongside the phylogenetic diversity of snake diets. Their study focused on North American pitvipers and used Shannon’s diversity index to quantify the diversity of both venoms and diets; they also used phylogenetic diversity of prey items to account for more divergent prey species being more likely to be physiologically different. They found the expected result of more generalist diets predicting more diverse venoms, but only with their phylogenetic diversity measure, suggesting that the degree of divergence between prey species is important for the evolution of target venoms. They argue that the phylogenetic diversity of prey likely predicts the evolution of venom complexity: snake venom potency decreases with increasing phylogenetic distance from natural prey, and the taxonomic breadth of the snake diet predicts the effectiveness of venom against a more diverse set of prey. This relationship between diet diversity and venom complexity implies parallel evolution of complexity levels within several independent venom gene families. However, interestingly they found that the relationship held only for three of the four major toxin families in viper venoms and was strongest in those typically most abundant in the venoms of viperid snakes (snake venom metalloproteinases and serine proteases). While this clearly demonstrates, for this group of snakes at least, that diet diversity does indeed favour toxin diversity in a manner related to the physiological diversity of prey, it also suggests that considering relative abundance and evenness as a component of diversity may give insights into how this relationship operates.

Herein, we quantify the diversity of snake diet and venom composition with the aim of understanding whether and how toxinological diversity is associated with dietary diversity. In doing so, we build upon previous work, notably [20], by (1) using published venomic data with relative abundances to generate a wider taxonomic sampling of snakes beyond a single geographic area and subfamily and (2) considering two different diversity metrics to try to disentangle effects of richness and evenness of venom toxins.

## 2. Results

Our dataset includes 178/193 (for family/order level diet analyses) individual venomic studies across 61/66 species (intraspecific variation was explicitly accounted for in our analyses), 35 genera, and five families of caenophidian snakes: Viperidae, Elapidae, Colubridae, Dipsadidae, and Homalopsidae. Using Simpson’s index, we found that snakes with more diverse diets at both Family- and Order-level had more diverse venom composition (Family: χ^2^ = 43.427, *p* = 4.402 × 10^−11^; Order: χ^2^ = 7.087, *p* = 0.008; Table 1; Figure 1). In contrast, using Shannon’s index, we found the opposite pattern, wherein snakes with less diverse diets had more diverse venom composition (Family: χ^2^ = 101.596, *p* < 2.2 × 10^−16^; Order: χ^2^ = 57.847, *p* = 2.831 × 10^−14^; Table 1; Figure 2).

The regression coefficients for diet diversity as a predictor of venom diversity are very similar when a given diversity index is calculated for prey at either Family or Order level (Table 1). This suggests that for those two taxonomic scales, at least, the same relationship holds between diet and venom diversity at equivalent magnitudes.

Although we included both transcriptomes and proteomes in our venom composition data, we note that previous work has found that these are strongly correlated in snake venom (e.g., [20,21]), and we explicitly accounted for variation between studies in our analyses which will partially control for this. It is also unlikely that the venomic method used was biased with respect to our question. For instance, there is no obvious reason why proteomes would be mostly generated for diet generalists and transcriptomes for diet specialists or vice versa. Moreover, *t*-tests on our dataset were unable to detect any difference between proteome and transcriptome samples in venom diversity measured either as the raw number of protein families (t = −1.096, df = 38.826, *p* = 0.280), Simpson’s diversity index (t = 1.763, df = 48.730, *p* = 0.084), or Shannon’s index (t = −0.162, df = 46.064, *p* = 0.872). Hence, on methodological, a priori, and empirical grounds, our results are very unlikely to be biased by any heterogeneity in toxin diversity measures from transcriptomes versus proteomes.

## 3. Discussion

We found contrasting relationships between diet diversity and toxin diversity of snake venoms depending on the diversity index used, with more diverse diets being associated with greater venom diversity as measured by Simpson’s diversity index but with lower venom diversity by Shannon’s index. These results are robust to the taxonomic scale at which diet diversity was measured, are based on a taxonomically broad sample, and were obtained while accounting for both the influence of phylogeny and variation in intraspecific venom diversity between venomic studies. Given the key differences between Shannon’s and Simpson’s diversity indices, our results suggest that high snake venom toxin diversity is indeed related to more generalist diets but that this effect is driven by variation in the relative abundance of toxin families rather than by more toxin families being present. Specifically, venoms from snakes with more generalist diets do not have more different toxin families, perhaps even slightly fewer, but they tend to have a more even spread of toxin families such that they are less dominated by one or few toxin families. Importantly, because our study was focused on the level of toxin families, we were unable to account for the diversity of isoforms within these families, leaving the possibility that low Simpson’s diversity (venoms dominated by few abundant toxin families) is associated with high diversity within toxin families. In fact, the most abundant toxin families in snake venoms also tend to be the ones that have faster evolutionary rates and hence more isotypic diversity [21], such that the abundance of toxin families could be a reasonable proxy of isoform diversity within those families.

To our knowledge our study is the first to use Simpson’s diversity index to investigate venom complexity, but previous work gives some (variably direct) comparisons with our results from Shannon’s index. In cone snails, although intraspecific variation in *Conus ebraeus* finds more toxin genes in populations with more diverse diets [17], no such correlation was found in an interspecific study [22]. However, in the latter study, the non-significant trend appears to be negative (i.e., in the same direction as we find) and may be underpowered by a comparison of only six data points, as with a larger sample the same study found that cone snails with a venom gland had lower diet diversity than those without [22]. Hence, evidence in cone snails is variable when diet diversity is measured with Shannon’s index, but that from interspecific comparisons is at least consistent with our results of lower venom complexity in species with more generalist diets.

In snakes, Holding et al. [20] used a dense and robust sampling of North American pitvipers and Shannon’s index for measuring the diversity of both diet and venom complexity. Despite also finding a slight trend in the same (negative) direction as our results and the cone snail study above [22], this was non-significant and explained very little of the data. The lack of evidence here is unlikely to be merely a result of sample size-induced low power, as sampling was good, but perhaps our broader taxonomic sample gave more variation between our species to detect an effect than may be available within one clade of a single subfamily. Although no formal statistical analyses were conducted, a study of the venom composition of 28 Australian elapid snakes [7] also found results consistent with our study, allowing that number of toxin classes as a measure of diversity is more closely linked to Shannon’s index than Simpson’s. Specifically, they noted that those species with the most complex venom (in terms of toxin classes) were not the snakes with the most generalist diets but those with relatively specialized diets [7]. In any case, while it is unusual to find good evidence of an effect of lower (Shannon) diversity venom in species with more generalist diets, this result is not inconsistent with previous work using comparable measures.

Our negative relationship between diet and venom diversity under Shannon’s index might have some precedent in the literature, but it remains difficult to explain. Holding et al. [20] explained their lack of evidence for any relationship with this measure as capturing the ‘wrong’ information: the phylogenetic (and thus presumably physiological) divergence of the prey was more important than straightforward taxonomic diversity. Since they calculated Shannon’s diversity measure for diet at the level of species, where substantial diversity might arise from several very closely related species, this might explain why we found an effect more consistent with their phylogenetic diversity measure as we considered the diversity of prey at the family and order levels. Therefore our results likely capture greater phylogenetic diversity in the diet but perhaps not sufficiently to relate to substantial physiological variation. Maybe several families of rodents or passerines (for instance) are sufficiently similar to avoid the need for many diverse venom toxins (though we note we obtain essentially the same results using order-level diversity). Jackson et al. [7] instead propose that streamlined venoms in dietary generalists may be the result of finding a ‘good’ set of toxins which act on diverse prey groups, are difficult to evolve resistance to and economise on toxin production by reducing the diversity of other components of their venom. This would certainly explain why some generalist snakes have relatively simple venoms, but it is difficult to see why the same argument would not apply to more specialist snakes also, particularly as a ‘good’ toxin by this measure should be easier to find as it needs to work on less prey. We note that Jackson et al. [7] did identify streamlined venoms amongst the specialised members of plesiomorphic genera of Australasian elapid and hypothesised a complex scenario in which streamlined venoms were first complexified via the recruitment of additional toxin families occurring in parallel with shifts towards more generalised diets. Subsequently, venoms were ‘consolidated’ around key toxin families, leading to secondary streamlining. They suggested that the secondarily streamlined venoms, in terms of toxin families, might be characterised by higher diversification of toxin isoforms within those families: fewer toxin classes but each containing more diversity within. Assuming abundance of a family is proportional to its isoform diversification, as per [21], this would predict lower diversity measured via both Shannon’s and Simpson’s diversity indices, but we find these show opposing trends. Such contrasting results are suggestive of complexities in the evolution of venom diversity that are beyond the scope of the present analysis (see [7] for a more detailed discussion of some of these considerations). Hence, although somewhat foreshadowed, the negative relationship between Shannon’s diversity of diet and venom remains to be convincingly explained, and so suggests itself as an interesting avenue for future researchers.

One explanation might be apparent by considering diversity in the wider context, as enabled by our use of two distinct diversity metrics. In addition to the different weights given to richness versus evenness/abundance, the Shannon diversity index responds most strongly to changes in the importance of the rarest species, while the Simpson index responds most strongly to changes in the proportional abundance of the most common species [23]. So, according to our results taken holistically, a more generalist diet should be associated with venom with toxins from different protein families but with fewer rare (or disproportionately abundant) toxin families than the venoms of species with more specialist diets. Put another way, a diverse diet might lead to a venom with a more even composition of its (perhaps fewer) toxin types, whereas a specialist diet might lead to a venom with more toxin families but dominated by a few abundant families (which may exhibit considerable diversity in isoforms). This would be consistent with two strategies for venom evolution: equal ‘firepower’ to take down a wide range of prey or a few dominant effects targeted at a few prey species and a range of more ‘subordinate’ (in terms of relative abundance) toxins. An intriguing corollary of this latter strategy is that the rarer toxins may be relatively unimportant in the venom and hence subject to more relaxed selection. One test of these ideas would be molecular evolutionary estimates of selection on toxin classes with two main predictions. First, that selection pressure should decrease in tandem with relative abundance in the venom, and second, that the variation of selection pressure between toxin classes would be lower for more dietary generalists. There is already some support for the first of these predictions [21], but as that potentially has a wider range of explanations than the second prediction, testing of the latter is crucial.

Our finding that Simpson’s diversity index positively scales with venom complexity is more novel but also more intuitive to explain, assuming that greater toxin diversity corresponds to a greater diversity of physiological targets that can be attacked. Essentially, dietary specialists should have venoms more dominated by a few components, presumably those most effective on the relatively small number of prey types that need to be subjugated. In contrast, dietary generalists should have a need to maintain as high a level of as many different types of toxins as possible, to account for the defences and/or diverse physiologies of the wide range of prey they need to subjugate. The optimal way to do this is to have a more even distribution of toxin abundances since an increase in the relative abundance of any one toxin class necessitates a decrease in others. This straightforward interpretation explains several previous results, including the low complexity venoms of (relatively specialist) sea snakes [24], the finding that diet breadth-toxicity relationships are driven by specialists (because those are more likely to have more dominant prey-specific toxins) [16], links between more diverse prey classes and more diverse venom activities (assuming toxicological activity is linked to sufficient quantities of several different toxins) [18], and ontogenetic shifts from specialist to generalist feeding ecologies being accompanied by an increase in toxin class diversity [7].

In several comparative venomics studies, there are examples of difficult-to-explain deviations from a typical pattern found in closely related species. One example is the mambas (genus *Dendroaspis*), for which four ‘green’ mamba (sub)species have broadly similar compositions, but the black mamba (*Dendroaspis polylepis*) has a notably different venom [25]. This has been explained loosely in the context of diet, specifically that black mambas are more terrestrial than green mambas, and they feed more on mammals which might be more dangerous and need specialist venoms [25]. While this is almost certainly correct, it does not explain how their venom should differ, just that it should be different as a result of different diets. In the context of our results, it is notable that a major way in which the black mamba’s venom differs from its congeners is in a reduced dominance by three-finger toxins (3FTx) and a more even split at least between the two main components (3FTx and Kunitz-type toxins); in essence, it seems likely that black mamba venom will have higher Simpson’s diversity than other mambas. Although good data on mamba diets are limited, several prey items have been reported for black mambas, Jameson’s mamba (*D. j. jamesoni*), and eastern green mambas (*D. angusticeps*) [26,27]. Taking those prey items that can be identified at family and order level (to match our data categorisation), black mambas were reported to feed on eleven families in nine orders, Jameson’s mambas on nine families in eight orders, and eastern green mambas on seven families in four orders. In other words, shifting towards a more diverse diet with less focus on particular groups (e.g., Passeriformes) than green mambas would be predicted to lead to changes in venom composition along the lines of what we see in the unusual venom of the black mamba. Hence, our results offer novel ways of interpreting venom compositions that diverge from those typical of their close relatives.

The recent increase in consideration of the importance of diversity *per se* in the relationships between diet and venom evolution in snakes has strong potential to provide new insights into the selection pressures imposed by diet on venom composition and hence, give a greater understanding of the ecology and evolution of snake venoms in general. Previously, sufficient data on both venoms and diet were lacking on a scale that enables comparative analyses such as ours. Dietary information was typically scattered in isolated and sometimes obscure sources, and summaries (such as from field guides) are low resolution and potentially based on little or no direct evidence. The recent availability of databases of diet records (e.g., [26]) brings together such sources in a way that has changed what questions can be asked and how. Similarly, venom composition data was patchy even for those species considered ‘well-studied’ until the advent of venomics protocols which have the potential to provide comprehensive data on many species. Vitally important in this latter vein is the production of quantitative venomics data since the relative abundance of different components provides so much more detail than simple lists of proteins found, and (as in the case here) this data may frequently be highly biologically meaningful. Similarly, the availability of ‘locus-resolved’ venomics data may make it possible to differentiate between levels of diversity between and within toxin classes, which may be a crucial consideration in future studies. Indeed, some of the predictions that could be derived from both our current results and previous work (e.g., [7]) require measures of isoform diversity within toxin families, so this is likely to be a fruitful area of research to understand the evolution of venom diversity.

We note that our dataset was not able to link the diet and venom data at an individual level, wherein diet data were collected from the same individual snakes as the venom data. Intraspecific variation in snake venom (and diet) has been widely recognized (e.g., [5]), and venomic data may be obtained from pooled venoms or venom from an individual snake. Hence, a limitation of our study is that the individual snakes contributing to the diet data are not those contributing to the venomic data. Nevertheless, we highlight that we explicitly incorporated intraspecific venom variation in our study, using data from multiple venomic studies for a given species where available and models which consider such variation directly. The scope of our study was on interspecific relationships between venom complexity and diet diversity, such that by controlling for intraspecific variation, we consider macroevolutionary patterns of venom evolution rather than relationships between populations within a species. Although intraspecific studies of venom variation are vital to our understanding of venom evolution [2,28], our study has a different focus and cannot address questions related to how individual snake venoms are related to the diets of individual snakes.

We have presented evidence that toxinological diversity of snake venoms is associated with diet diversity, but we have shown that this is largely via changes in abundance rather than the richness of toxins. Specifically, we find that generalist diets tend to favour venoms which are less dominated by a few highly abundant toxins families (compared to specialists) but have a more even composition even if fewer toxin families are present overall. Aside from providing a more detailed understanding of how diet and snake venom composition are related, our results also shed light on the origination of the infamous ‘complex cocktail’. The complexity in species with more specialist diets may involve the presence of more toxin classes, but most of those will be in very low abundance, and emphasis in the venom is put on a few highly abundant toxin families (which may exhibit a high diversity of isoforms within them). In contrast, the complexity in dietary generalists results from (perhaps fewer) toxin families which are present in more equal amounts, leading to potentially higher quantities on average of any given toxin family (since fewer will be rare in the venom) and hence potentially more complex symptoms from envenomations.

## 4. Materials and Methods

We collated a taxonomically broad database of venom composition via an extensive literature review of snake venomics studies. Venom composition data were collected from 164 studies (Appendix A), and their concordance with the non-exhaustive online Database of Tropical Pharmacology was checked (6 May 2022) [29]. We, therefore, collected all papers containing quantitative proteomic or transcriptomic studies reporting a proportional composition of venoms. For each transcriptomic or proteomic profile found, we recorded the type of analysis (proteome vs. transcriptome), name of the species, and relative abundance for each protein family identified in the study.

We used Squamatabase [30] in R 4.2.0 [31] to collect diet information for species we were able to include in our venom composition dataset. Squamatabase is an open-source R package and curated database of predator–prey records for snakes based on published diet records. We recorded prey items consumed by family and order, including both the identity of the prey (at those taxonomic scales) and the number of records for each prey type. We did not use species-level prey identification because (1) this level of information was often not available in the data, and (2) relationships between diet and venom diversity are expected to operate via diversity in physiological targets of toxins [7], so higher taxonomic levels are likely to capture more of this than different (potentially closely-related) species. Hence, our higher taxonomic classifications are likely a more relevant level for our questions.

A dated phylogeny for snake species in our dataset was obtained from the posterior sample of trees from [32] accessed via the VertLife database (http://vertlife.org/ (accessed on 14 March 2022)). We downloaded 1000 trees matching our species sampling and calculated the maximum clade credibility tree in phangorn 2.7.1 [33], which was then used in our analyses.

Our initial dataset contained 356 individual records belonging to 173 species (Appendix A), but we then undertook a rigorous data cleaning strategy to ensure as high-quality data as possible. For the venomic data, we removed incorrect records: for instance, where the total relative (per cent) abundance was above 110%, with some leeway of about 100% to allow for rounding errors. For the diet data, we removed all species for which we have less than five prey items recorded for our family and ordered datasets separately to ensure our measures of diet diversity were not unduly influenced by isolated (potentially unusual) records. After cleaning the data, our final datasets included 193 individuals belonging to 66 species for order-level diet categories and 178 individuals belonging to 61 species for family-level diet categories (Appendix A).

We calculated Shannon’s and Simpson’s diversity indices to quantify the diversity of both venom and diet for each record. These are standard indices for biodiversity, more commonly applied to the diversity of species in ecology, which incorporate both the number of different components (commonly species richness, but prey types or toxin families in our study) and the abundance of each component as two key aspects of diversity [19]. Importantly, the two indices differ in their emphasis on richness versus abundance; Simpson’s diversity index gives more weight to the abundance of components (so higher values when diversity is more evenly spread and lower values when composition is dominated by a few components), whereas Shannon’s index gives relatively more weight to richness (so more closely represents what we might intuitively think of as ‘diversity’). These differences enable us to gain a deeper understanding of *how* diet diversity and venom complexity are related, with a focus on distinguishing the major effects of the two components of biodiversity.

For direct reference, the equations for the diversity indices used are provided here. Shannon’s diversity index is calculated as follows:$$Shannon\;index=-\sum_{i=1}^{R}p_{i}ln(p_{i})$$
where $p_{i}$ is the proportion of the diet or venom comprised of the *i*th prey type or toxin family, and *R* is the total number of prey types or toxin families in the diet or venom.

Simpson’s diversity index is calculated as follows (symbols as described above):$$Simpson\;diversity\;index=1-\sum_{i=1}^{R}p_{i}^{2}$$

All analyses were conducted in R 4.2.0 [31], with basic handling of the phylogenetic trees, data handling, and plotting using the packages ape 5.3 [34], geiger 2.0.7 [35], caper 1.0.1 [36], and ggplot2 3.3.5 [37].

Because we had multiple data points for a given species, we tested for an effect of diet diversity on venom diversity using Ives et al.’s [38] phylogenetic regression, which incorporates intraspecific variation. We ran four versions of this model in phytools 0.7.90 [39] using either Simpson’s diversity index or the Shannon index as the measure of diversity, and each of these two variants with either family or order level categorisation of prey. The *p*-values were obtained using a likelihood ratio test between each model and one using the same data but enforcing no effect of diet diversity on venom diversity (i.e., constraining the regression coefficient to be 0, creating an ‘intercept only’ model). Because model convergence can be sensitive to starting conditions in this model in cases where parameters are near the estimation boundaries (i.e., with near-zero variances, though unlikely to be the case in our data), we ran each analysis 100 times and report the mean estimate in our results.

## Figures and Tables

**Figure 1 toxins-15-00251-f001:**
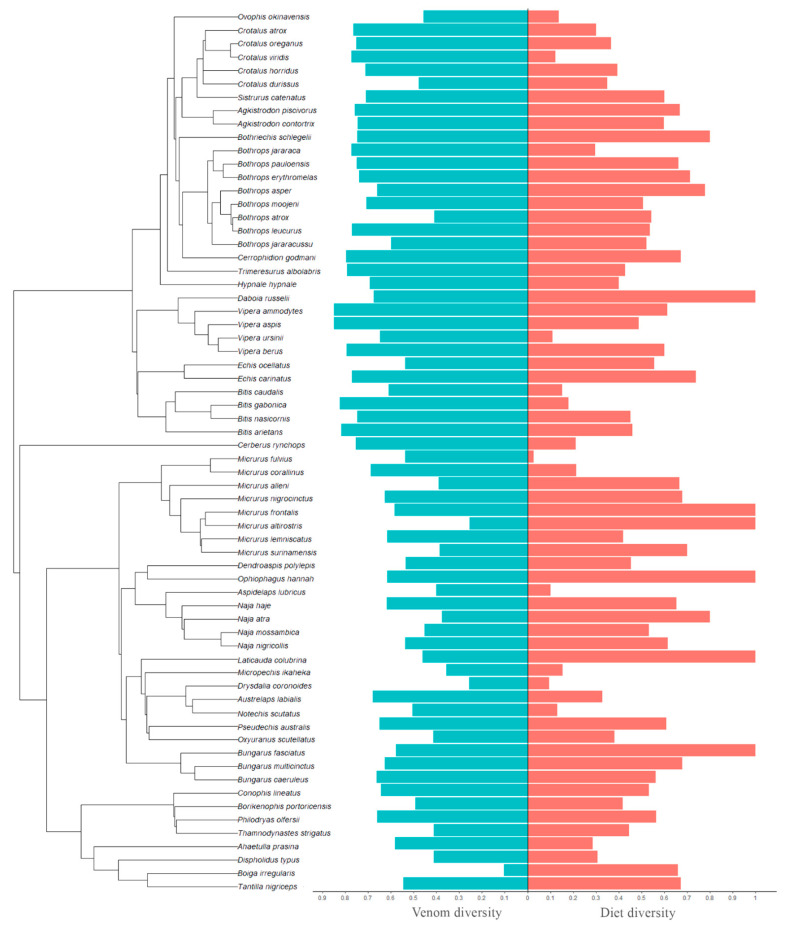
Phylogeny of all species included in our study. Bars next to the tips of phylogeny indicate mean diet (red) and venom (blue) diversity using Simpson’s diversity index.

**Figure 2 toxins-15-00251-f002:**
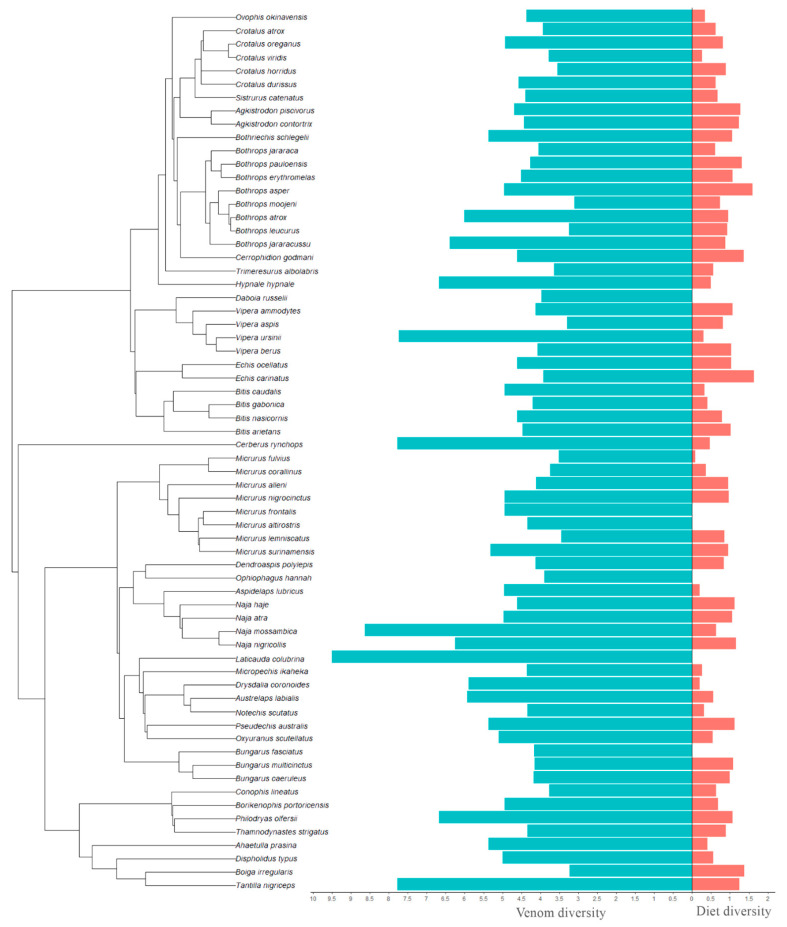
Phylogeny of all species included in our study. Bars next to the tips of phylogeny indicate mean diet (red) and venom (blue) diversity using Shannon’s diversity index.

**Table 1 toxins-15-00251-t001:** Model outputs for pGLS regressions incorporating intraspecific variation for both Simpson and Shannon diversity indices calculated at two taxonomic levels for diet (F: Family-level, O: Order-level). A likelihood-ratio test (LRT) was performed between a pGLS model with the regression coefficient estimated (pGLS) and one with the coefficient fixed to 0 (i.e., assuming there is no effect of diet; pGLS (intercept only)). Estimated parameters are given for the full model. Index = version of the index and taxonomic level of diet used for each model; logLik = log-likelihood for each model; LR = likelihood-ratio; *p* = *p*-value for the LRT.

	pGLS			pGLS (Intercept Only)		LRT
Index	logLik	Coefficient	Intercept	logLik	LR	*p*
Simpson (F)	−28.866	0.137	0.035	−50.580	43.427	**4.402 × 10^−11^**
Shannon (F)	−176.937	−0.183	2.748	−227.735	101.596	**<2.2 × 10^−16^**
Simpson (O)	−24.361	0.124	0.045	−27.905	7.087	**0.008**
Shannon (O)	−228.205	−0.186	2.736	−199.282	57.847	**2.831 × 10^−14^**

## Data Availability

Data supporting reported results can be found at: https://doi.org/10.6084/m9.figshare.22179949.v1.

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
