# Peer review of "Diversity Begets Diversity When Diet Drives Snake Venom Evolution, but Evenness Rather Than Richness Is What Counts"

_toxins, 2023, doi:10.3390/toxins15040251_

Round 1

Reviewer 1 Report

This paper analysis how venom composition relates to diet in snakes with a particular focus on the role of both the richness/evenness of diet relates to the richness/evenness of a venoms composition. The idea is clear simple and a much needed analysis at the scale at which the authors test the idea. Overall, I think the authors do a good job at collating the necessary data and outlining the results, even if some of them are unexpected. Overall I like the paper but would like to see just a little bit more exploration in the analysis as noted below.

My main recommendations are in relation to a lack of analysis the phylogenetic diversity of the diet and the possibility of teasing out the role of evenness in the analysis. In relation to the phylogenetic diversity of the diet, as the authors mention this has been found to be an important driver of venom composition diversity in Holding et al 2021. While the data is not at the species level the phylogenetic diversity of the data could still be calculated at higher taxonomic levels using something like Faiths phylogenetic distance. This, or some other course measure of diet phylogenetic diversity, might help tease out the effect relating to richness found here. Similar to this, while the authors note that differences between Shannon and Simpsons diversity indexes are likely due to weighting of evenness there is no further exploration of this. An additional analysis using a metric that just measures evenness, such as taking the slope from a rank abundance curve, would likely offer further insights into the strength of evenness in explaining these results.

Additional comments.

While the figures 1 and 2 are good for giving the reader an idea of the overall variance in the data, I would like to the see a scatter plot of the relationships between venom and diet diversity as it is difficult to visually capture relationships using bar charts.

Lyons et al 2020 is included twice in the reference section and referred to as reference [10] and [16] in the main text. Please review, this could be that Lyons should only be included once or that the other study testing the loss of eggs in the diet, Healy et al 2019, was meant to be included when [10] was referenced.

Reviewer 2 Report

Venoms are diverse samples composed by multifunctional toxins. This molecular and functional diversity leads to complex and variable clinical manifestations. As a result, the treatment of snakebite envenomation is a big challenge. In this sense, the understanding of mechanisms underlying venom variability is a crucial. Here, the authors explored the link between diet and venom composition. The objective is clear and relevant. The manuscript is well-written and presents novel and interesting findings. However, the introduction is quite long, in contrast to the results section, which is quite brief in the present version.

1. The introduction is long with many examples. These, if relevant in the context of this study, should be presented in the discussion in comparison with the main findings.

2. Line 20: Why is this question mark used here?

3. Some sentences must be supported by references. For example, lines 36-41, 42, 98-101.

4. I expected a comparison of venom diversity indices between snake families. Why was this comparison not performed?

5. The quality of the figures needs to be improved. Species names and numbers are blurred.

6. More detail on how the proteomic data were used in calculating the indices would be appropriate. I encourage authors to detail the calculation, with formulas and variables. The findings are interesting. From the biochemical point of view, elapid venoms are composed of few families of proteins. For example, corals basically feature PLA2 and 3FTs, along with a few other families. In contrast to viper venoms, where a greater number of toxin families have been described. However, this is not reflected in the analysis presented by the authors in figures 1 and 2. This is very interesting, and I would like to read the discussion and author`s views.

7. Limitations of the studies were not discussed. Just the lack of direct connection between proteomic and dietary data limited the scope of the study, although it did not reduce the scope. I consider the strategy used by the authors timely, however this limitation must be discussed. Proteomic results can be obtained from pools of venoms or venom from an individual. Intraspecific venom variability has been extensively reported. Therefore, the diet data does not necessarily correspond to the snake or snakes used in the proteomic study.

Round 2

Reviewer 2 Report

I recommend the publication of this manuscript in its current version.